# Variable Expressivity and Allelic Heterogeneity in Type 2 Familial Partial Lipodystrophy: The p.(Thr528Met) LMNA Variant

**DOI:** 10.3390/jcm10071497

**Published:** 2021-04-03

**Authors:** David Araújo-Vilar, Antía Fernández-Pombo, Berta Victoria, Adrián Mosquera-Orgueira, Silvia Cobelo-Gómez, Ana Castro-Pais, Álvaro Hermida-Ameijeiras, Lourdes Loidi, Sofía Sánchez-Iglesias

**Affiliations:** 1UETeM-Molecular Pathology Group, Department of Psychiatry, Radiology, Public Health, Nursing and Medicine, IDIS-CIMUS, University of Santiago de Compostela, 15782 Santiago de Compostela, Spain; david.araujo@usc.es (D.A.-V.); antiafpombo@gmail.com (A.F.-P.); silviacobelog@gmail.com (S.C.-G.); alvaro.hermida@usc.es (Á.H.-A.); 2Division of Endocrinology and Nutrition, University Clinical Hospital of Santiago de Compostela, 15706 Santiago de Compostela, Spain; anaisabel0121@gmail.com; 3Burnett School of Biomedical Sciences, College of Medicine, University of Central Florida, Orlando, FL 32827, USA; berta.victoriamartinez@ucf.edu; 4Department of Hematology, University Clinical Hospital of Santiago de Compostela, Santiago de Compostela, 15706 Santiago de Compostela, Spain; adrian.mosquera@live.com; 5CIBER Fisiopatología de la Obesidad y la Nutrición (CIBERobn), 28029 Madrid, Spain; 6Division of Internal Medicine, University Clinical Hospital of Santiago de Compostela, 15706 Santiago de Compostela, Spain; 7Fundación Galega de Medicina Xenómica, 15706 Santiago de Compostela, Spain; lourdes.loidi.fernandez@sergas.es

**Keywords:** type 2 familial partial lipodystrophy, FPLD2, *LMNA*, T528M

## Abstract

Type 2 familial partial lipodystrophy, or Dunnigan disease, is a metabolic disorder characterized by abnormal subcutaneous adipose tissue distribution. This rare condition results from variants principally affecting exons 8 and 11 of the *LMNA* gene. In this study, five FPLD2-diagnosed patients carrying the c.1583C>T, p.(Thr528Met) variant in exon 9 of the *LMNA* gene and with obvious clinical heterogeneity were evaluated. Specific polymorphisms in *LMNA* and in *PPARG* were also detected. Exhaustive clinical course, physical examination, biochemical features and family history were recorded, along with the assessment of anthropometric features and body composition by dual-energy X-ray absorptiometry. Preadipocytes obtained from a T528M patient were treated with the classic adipose differentiation medium with pioglitazone. Various adipogenes were evaluated by real-time PCR, and immunofluorescence was used to study intracellular localization of emerin, lamin A and its precursors. As demonstrated with Oil red O staining, the preadipocytes of the T528M patient failed to differentiate, the expression of various adipogenic genes was reduced in the lipodystrophic patient and immunofluorescence studies showed an accumulation of farnesylated prelamin A in T528M cells. We conclude that the T528M variant in *LMNA* could lead to FPLD2, as the adipogenic machinery is compromised.

## 1. Introduction

Familial partial lipodystrophies are a group of Mendelian diseases characterized by the loss of subcutaneous adipose tissue in the lower limbs and sometimes also in the upper limbs, with abnormal fat accumulation in other regions. Affected subjects have insulin resistance leading to complications such as diabetes, dyslipidemia, liver steatosis and increased cardiovascular risk. FPLD2 patients exhibit lipoatrophy in the upper and lower limbs and buttocks from puberty, particularly in women, and accumulation of fat in the neck, face and visceral depots, following an autosomal dominant pattern of inheritance.

FPLD2 results from variants in the *LMNA* gene (1q21–q23), which codes for type-A lamins by alternative splicing: lamin A, lamin C, lamin C2 and lamin A∆10. These are intermediate filament proteins, forming polymers at the nuclear lamina, a meshwork underlying the inner nuclear membrane. They exert recognized functions in a range of transcendental biological processes: mechanical function in nuclear shape maintenance, conservation of nuclear and chromatin architecture, DNA replication and transcription, cell cycle and cellular senescence/apoptosis, cell proliferation, tumor progression and interactions with other nuclear and cytoplasmatic proteins (thus regulating the communication between these two compartments).

Type-A lamins have a characteristic structure: a small N-terminal head domain, a coiled-coil rod domain divided into four α-helix segments, and a globular C-terminal IgG-like end or tail domain. The rod domain allows coiled-coil dimerization, while the head and the tail are involved in the end-to-end assembly of the polymer, and greater associations. The main isoforms of type-A lamins are lamins A and C, both identical to codon 566, from which lamin C lacks part of the C-terminal region, including some amino acids of exon 10, exon 11 and exon 12. It is important to note that FPLD is generally caused by heterozygous amino acid changes in the C-terminal domain of lamin A/C, usually between exons 8 and 11 (>90% of variants affect codon 482 of the gene, a mutational hot spot). These regions codify for the protein elements involved in prelamin A processing to generate mature lamin A.

Previous studies have suggested that variant c.1583C>T, p.(Thr528Met) only produces pathological manifestations when it appears in compound heterozygosity [1,2]. We report here the cases of five patients from four pedigrees with clinical diagnosis of FPLD2, carrying the p.(Thr528Met) missense variant in exon 9 of the *LMNA* gene. Various genes related to other forms of familial partial lipodystrophies were also analyzed by NGS (next-generation sequencing). The primary objective of this study was to clearly establish that this *LMNA* variant is the primary cause of the lipodystrophy observed in our patients by evaluating whether it alters adipogenesis in human preadipocytes. No less important was the aim to examine the complex genotype-phenotype associations in more depth along with the wide clinical heterogeneity observed.

## 2. Experimental Section

This study was approved by the Ethics Review Panel of Xunta de Galicia, and carried out according to the ethical guidelines of the Helsinki Declaration. Patients gave informed consent for participation in the study and the publication of clinical, biochemical, and genetic information.

### 2.1. Subjects, Analysis and Interpretation of Variants

All patients studied are heterozygous for the variant c.1583C>T, p.(Thr528Met) in the *LMNA* gene. The search for variants in genes *AGPAT2*, *AKT2*, *BANF1*, *BLM*, *BSCL2*, *CAV1*, *CIDED*, *ERCC6*, *ERCC8*, *FBN1*, *KCNJ6*, *LIPE*, *PCYT1A*, *PIK3R1*, *PLIN1*, *POLD1*, *PPARG*, *PSMB8*, *PTRF*, *SPRTN*, *WRN* and *ZMPSTE24* was made by NGS (Ion torrent System, Thermo Fisher Scientific, Waltham, MA, USA) sequencing of the entire coding region of the genes and the flanking intronic regions. The capture of the regions of interest was performed using SureSelectXT Custom (Agilent, St Clara, CA, USA) and data analysis was performed using computer tools: TMAP 5.4.11, TVC 5.4–11, GATK v3.8–0, Picard 2.10.2-SNAPSHOT, BEDtools v2.26.0, SAMtools 1.5 and ExomeDepth 1.1.10. Interpretation of classification of variants was done by following the guidelines of the American College of Medical Genetics and Genomics and the Association for Molecular Pathology (ACMG, Bethesda, MD, USA) [3]. The classification of the variants identified reflects the current state of scientific knowledge and they might change as new scientific information becomes available. Different resources and databases were used for the variant classification as Varsome [4], gnomAD [5] (https://gnomad.broadinstitute.org, accessed 2 April 2021), ClinVar [6] and dbSNP [7].

### 2.2. Body Composition Studies

Height and body weight were measured by standard procedures. Skinfold thicknesses were measured with Lange Skinfold calipers (Cambridge Scientific Industries, Watertown, MA, USA) at two truncal (subscapular and suprailiac) and four peripheral sites (biceps, triceps, thigh and calf) on the right side of the body. The distribution of body fat was assessed via whole-body dual-energy X-ray absorptiometry (DXA), using a Lunar model DPX apparatus (GE Healthcare Lunar, Madison, WI, USA) [8].

### 2.3. Biochemical Analyses

Fasting serum samples were analyzed for glucose, total cholesterol, HDL-cholesterol, LDL-cholesterol, triglycerides, thyroid stimulating hormone (TSH), creatinine, and creatine kinase (CK) as described previously [9]. Glycated hemoglobin (HbA1c) was measured using ion-exchange high-performance liquid chromatography (Bio-Rad Laboratories Inc., Hercules, CA, USA). Aspartate aminotransferase (AST), alanine aminotransferase (ALT) and gamma-glutamyl transpeptidase (GGT) were determined with enzymatic methods on an ADVIA analyzer (Siemens, Bayer Diagnostics, Tarrytown, NY, USA). Plasma insulin concentrations were determined in duplicate by chemiluminescence, using a commercial kit (Nichols Institute, San Juan Capistrano, CA, USA). Plasma leptin levels and C-peptide were determined by ELISA assay (DRG International, Inc., Springfield, NJ, USA).

### 2.4. Adipose Tissue Biopsies and Cell Culture

A small sample of subcutaneous adipose tissue was obtained from the lower back area of Case #1 at 42 years of age. A control of normal adipose tissue sample was obtained from the back of a 32-year-old woman who underwent programmed surgery for lipoma extraction, in accordance with current Spanish legislation. Small pieces of adipose tissue were placed on a 60 mm dish (BD FalconTM; Mississauga, ON, Canada) containing Dulbecco’s modified Eagle’s medium (DMEM) plus 30% fetal bovine serum (FBS) and gentamicin 50 μg/mL, and incubated at 37 °C with 5% CO_2_ in a Water-Jacket CO_2_ incubator (NuAire; Plymouth, MN, USA). Preadipocytes were recognized by the presence of small lipid droplets in the fibroblast-like cells using a phase microscope. Subsequently, these preadipocytes were trypsinized (TrypLE™ Express Stable Trypsin-like Enzyme with Phenol Red; Gibco Life Technologies; Carlsbad, CA, USA) and cultured on 100 mm dishes in DMEM containing 10% FBS and penicillin-streptomycin 1%.

### 2.5. Adipocyte Differentiation Procedure

After confluence, preadipocytes were cultured on 35 mm multi-well dishes (6-well plates) in a differentiation cocktail containing Dulbecco’s modified Eagle’s medium plus 10% fetal bovine serum, insulin (1 µg/mL), dexamethasone (0.25 µM) and 3-isobutyl-1-methylxanthine (0.1 mM in DMSO) [10] for 3 days, with a *PPARG* agonist, pioglitazone (10 µM in DMSO; Alexis Biochemicals, Lausanne, Switzerland), after which this medium was changed for a growth medium containing 1 µg/mL insulin with pioglitazone (10 µM) for two more days. The cells were then left to differentiate for another 5 days with growth medium containing pioglitazone (10 µM) changed every other day. Non-differentiated preadipocytes were supplemented with equivalent concentrations of DMSO and used as controls.

### 2.6. Phase Contrast Microscopy

Cells were fixed in 10% formalin for 60 min at room temperature, washed three times with distilled water and then stained with 0.5% (*w/v*) Oil red O solution in 60% isopropanol for 60 min, at 22 °C. Cells were washed again three times with distilled water, and lipid accumulation was finally estimated by phase contrast microscopy.

### 2.7. RNA Extraction and Retrotranscription

Total RNA was extracted from preadipocytes, using TRIzol (Invitrogen, Madrid, Spain) as per the manufacturer’s instructions. RNA was reverse-transcribed by using M-MLV reverse transcriptase (Invitrogen) as previously described [11].

### 2.8. Real-Time PCR

Specific primers and probes designed by Universal Probe Library (Roche Diagnostics, Sant Cugat del Valles, Spain; Table 1) were used to determine the specific expression of *CEBPA*, *CEBPB*, *FABP4*, *GLUT4*, *LPL*, and *PPARG* and *PREF-1* genes in a Light Cycler 2.0 (Roche Diagnostics). Real-time PCR conditions are available upon request. Results were normalized for the internal control RNA polymerase II gene, using the 2-ΔΔ CT method [12].

### 2.9. Immunofluorescence

Cells grown on glass coverslips were fixed with 4% paraformaldehyde, at 4 °C, for 1 h, and permeabilized in 0.1% Triton X-100, at room temperature for, 10 min. After blocking for non-specific binding (4% BSA, 1 h, at room temperature), the coverslips were incubated in an appropriate primary antibody, at 4 °C, overnight (1:150 anti-prelamin A (ANT0045, Diatheva, Fano, Italy), 1:120 anti-farnesylated prelamin A (ANT0046, Diatheva, Fano, Italy), 1:50 anti-emerin (Novocastra Leica, Barcelona, Spain) and 1:100 anti-lamin A/C (N-18) (Santa Cruz Biotechnology, Heidelberg, Germany)). The ANT0045 does not bind carboxymethylated-farnesylated prelamin A, while no cross-reaction is observed between ANT0046 and full-length prelamin A [13]. The following day, after three washes, the coverslips were incubated for 1 h, at room temperature, in darkness, with 1:600 Cy2-AffiniPure F(ab’)2 Fragment and 1:600 Cy3-AffiniPure F(ab’)2 Fragment (Jackson Immunoresearch, West Grove, PA, USA) and counterstained with 1:1000 DAPI (Life Technologies, Madrid, Spain). Coverslips were mounted in Fluoromount medium (Sigma, Barcelona, Spain). Immunofluorescence staining was analyzed using an Olympus IX51 microscope (Olympus Corporation) equipped with an Olympus DP72 digital camera.

### 2.10. Statistical Analysis

Real-time PCR analyses were performed by triplicates. Statistical significance was determined by using a non-parametric Kruskal-Wallis test, followed by a Mann-Whitney U post hoc Bonferroni’s correction. Data are presented as mean ± standard deviation (SD), with statistical significance set at *p* < 0.05, and were evaluated by using SPSS for PC (release 22; SPSS, Chicago, IL, USA).

## 3. Results

Photographs of the studied subjects, pedigrees and color map of DXA scans are depicted in Figure 1, Figure 2 and Figure 3, respectively. Demographic, anthropometric, biochemical and clinical features are summarized in Table 2.

### 3.1. Case Reports

Case #1 is a 53-year-old female (Figure 1A). She has suffered from Crohn’s disease since her youth. The patient was diagnosed with FPLD when she was 42 years of age. The lipodystrophic phenotype started during childhood and was characterized by the lack of subcutaneous adipose tissue in limbs, abdomen and buttocks, rounded face, double chin, and a hypermuscular appearance with calf hypertrophy, phlebomegaly and small breasts. She did not have acanthosis nigricans, hirsutism nor cardiac disease (normal Holter and echocardiography). At that age, she had a normal BMI (23.7 kg/m^2^), and reduced skinfolds in limbs, but in normal range in trunk (Table 2). She had hepatomegaly, albeit with normal glucose metabolism and no insulin resistance. On the other hand, she was taking omega-3 fatty acids for hypertriglyceridemia and her plasma leptin levels were low. At the age of 49, she was diagnosed with breast cancer and high blood pressure. At the age of 52, she was diagnosed with diabetes mellitus, and at present she is on sitagliptin, fenofibrate, n-3 fatty acids and telmisartan. Regarding her relatives, her mother (deceased) and sister had a similar lipodystrophic phenotype. Moreover, her mother had diabetes, hypertriglyceridemia, eruptive xanthomata, ischemic cardiopathy and suffered a stroke. Her 59-year-old sister also had high blood pressure and was diagnosed with diabetes mellitus when she was 52 years old, but she never came to the consultation for a medical evaluation. However, her lipid profile was always normal. Both the proband and her sister carry a heterozygous transition of cytosine to thymine in codon 1583 (exon 9) of the *LMNA* gene, leading to a substitution of threonine for methionine in the highly conserved protein residue 528. It was not possible to sequence the DNA of the deceased parents.

Case #2 is a 58-year-old female diagnosed with diabetes mellitus at the age of 44 (Figure 1B). She was referred to the Endocrinology Division due to a lipodystrophic phenotype which began in childhood. She had fat loss in upper limbs, hips, thighs and calves, as well as fat accumulation in the face with double chin, in the upper back and the intra-abdominal region and scarce abdominal subcutaneous adipose panicle. Her upper and lower limbs were muscular with phlebomegaly. Her hands were large with thick fingers, and her calves were hypertrophied. There were no palpable lipomas. She presented acanthosis nigricans on the nape and axillae, and also some acrochordons. She had high blood pressure, and hypertriglyceridemia. Her menstruation was regular and did not suffer from fertility problems. She had no cardiovascular diseases. Abdominal ultrasonography showed hepatomegaly (10–12 cm) with heterogeneous echostructure and irregular margins, suggestive of non-alcoholic steatohepatitis and splenomegaly of 15 cm. At the age of 54, she was diagnosed with polyclonal hypergammaglobulinemia and mild thrombocytopenia secondary to hepatopathy. A gastroscopy performed at the age of 58 showed incipient esophageal varices. The patient referred that her deceased mother, her daughter and two sisters had a similar phenotype, suggestive of familial partial lipodystrophy. The mother and sisters had been diagnosed with diabetes mellitus. There was no ischemic heart disease in her relatives.

Case #3 is the 34-year-old daughter of Case #2. At 31 years of age, she presented an atypical, though not severe, FPLD phenotype with an androgenic distribution of fat but no obvious fat loss (Figure 1C). There was fat accumulation in the face, double chin and trunk (back and abdomen), but not in the hips, arms, buttocks or lower limbs. There was no hypermuscularity or well-defined muscles, except for minimal hypertrophy of the calves. The patient had no phlebomegaly. She claimed to have hyperphagia, and she had no diabetes mellitus, no dyslipidemia, no hepatomegaly and no atherosclerotic cardiovascular disease. At 34 years of age, she presented with hypertriglyceridemia and low fat in the arms, buttocks, hips, and lower limbs, and accumulation of fat in the face, double chin, trunk, abdomen, and axillae. She had a 2 cm lipoma on the right shoulder. She did not manifest hirsutism, and acanthosis was minimal in the axillae. Her palms and soles presented normal fat distribution, as did the rest of the physical examination.

Case #4 is a 20-year-old female who has presented an FPLD phenotype since adolescence, with fat accumulation in the face, double chin, dorsal region, axillae, neck, arms and scapular region (Figure 1D). The presence of fat was scarce in the hips, buttocks, thighs, and calves. Fat was preserved in the abdomen. The patient presented phlebomegaly and marked musculature but the hypertrophy of the calves was doubtful. The amount of fat in her upper limbs was normal with no hypermusculation or phlebomegaly. She had acanthosis nigricans in the nape, axillae and groin. No lipomas were present. She had hirsutism, and no alopecia. She had not developed diabetes mellitus or hypertension. Menarche was at age 12, and oligomenorrhea/amenorrhea occurred before starting contraceptive treatment. She suffered a pulmonary thromboembolism at age 19. There were no subjects with diabetes in her family. Her father has hypertriglyceridemia and his lower limbs have normal musculature. Her mother refused to be examined. A paternal aunt and a cousin exhibited a similar phenotype. There was a significant history of cancer in her family: Her paternal grandmother suffered from leukemia, a paternal cousin died at the age of 5 from leukemia, her paternal aunts suffered breast cancer and a paternal uncle died of lung cancer.

Case #5 is a 62-year-old female who has presented a classic FPLD phenotype since adolescence, with an absence of adipose tissue in the upper and lower limbs, buttocks and hips, as well as an accumulation of fat on the face, abdomen, chin, back and axillae (Figure 1E). Her musculature was well-defined in the arms, lower limbs and buttocks. She had minimal phlebomegaly in her arms, frontotemporal alopecia and hirsutism in her thighs. No lipomas or acanthosis were present. There was no hepatomegaly or splenomegaly. She had a multinodular goiter. The patient did not suffer from diabetes mellitus, but she did have hypertension, hyperlipidemia and muscle aches. At age 53, she underwent a bypass for a revascularized heart disease. She had no fertility problems. She claimed to have hyperphagia. Her father (deceased) exhibited a similar phenotype. He had no diabetes mellitus nor heart disease. Her paternal grandmother also had a similar phenotype.

The five cases studied are heterozygous for the variant c.1583C>T, p.(Thr528Met) in the *LMNA* gene (rs57629361, ENST00000368300.8, NM_170707.2(LMNA):c.1583C>T). This variant is classified as pathogenic according to ACMG guidelines (PM1, PM2, PM5, PP2 y PP3) and its allele frequency in the European Non-Finnish (ENF) population is 0.0021% (gnomAD). This variant was submitted just two times in ClinVar classified as VUS (variant of uncertain significance), one of them without specific associated condition in 2012 and the second one in 2020 associated to cardiomyopathy. In addition, Case #4 bears another variant in compound heterozygosis: c.[1583C>T];[1718C>T], p.(Thr528Met); (Ser573Leu), the first variant inherited from her father and the second one from her mother. The NM_170707.2(LMNA):c.1718C>T variant is classified as likely pathogenic (PM1, PM2, PP2, PP3 and PP5). Its allele frequency in ENF population is 0.012% (gnomAD). This variant was reported nine times in ClinVar with classification ranging from benign to pathogenic. Cases #2 and #3 are also heterozygous for the two rare variants: c.1495A>G, p.(Arg499Gly) (chr8:30945355, transcript ENST00000298139.7) in the *WRN* gene, and c.813C>G, p.(Lys271Asn) (chr15:91295027, transcript ENST00000355112.8) in the *BLM* gene. The NM_000553.5(WRN):c.1495A>G variant is classified as VUS (PM2, BP4). This rare variant was not found in any of the consulted databases (gnomAD and dbSNP) although it was submitted once in ClinVar as a VUS in Werner syndrome. The NM_000057.4(BLM):c.813C>G variant is also classified as VUS (PM2, BP4). It is neither found in genomic variant population databases consulted (dbSNP, gnomAD) nor in Clinvar. Moreover, Case #1 is heterozygous for the non-synonymous polymorphism c.34C>G, p.(Pro12Ala) in the *PPARG* gene (rs1801282, transcript ENST00000287820.10). The NM_015869.4(PPARG):c.34C>G variant is classified as benign (PP2, BA1, BP4, BP6) and its allele frequency in ENF population is 12.45% (gnomAD).

### 3.2. Cellular and Gene Expression Studies

Only partial differentiation was observed in control preadipocytes. These cells showed no staining with Oil red O at basal level, although some cell clusters with lipid droplets stained with Oil red O were evident after differentiation cocktail administration. However, the preadipocytes obtained from Case #1 remained uncolored, failing to differentiate (Figure 4).

Relative expressions of different genes involved in adipogenesis: *CEBPA*, *CEBPB*, *FABP4*, *GLUT4*, *LPL* and *PPARG* (Figure 5a–f), as well as *PREF-1* as a preadipocyte marker (Figure 5g), were quantified by qPCR. As expected, the expression of *PREF-1* was negligible in the fibroblasts culture (Figure 5g). The relative expressions of *CEBPA*, *LPL* and *FABP4* were significantly reduced for the T528M preadipocytes when compared with the control preadipocytes (−76%, −59% and −95%, respectively). The relative expression of *PPARG* was significantly increased in T528M cells (+230%) when compared to control cells. Differences of expression for *GLUT4* and *CEBPB* were not significant when comparing T528M cells with control cells. After the differentiation process, there were significant differences in almost all the studied genes when comparing the control preadipocytes with the treated cells: *PPARG*, *CEBPA*, *GLUT4*, *LPL* and *FABP4* were significantly increased (+27%, +133%, +37%, +18421% and +1320%), whereas *CEBPB* was decreased (−22%). When comparing the relative gene expressions after treatment between control and T528M, there was an obvious decrease in T528M cells for *CEBPA* (−87%), *LPL* (−100%) and *FABP4* (−96%). *PPARG* gene expression was significantly increased in T528M adipocytes (+190%).

Furthermore, the immunofluorescence staining of anti-prelamin A, anti-prelamin A cleaved farnesylated, anti-emerin and anti-lamin A/C was compared between control and T528M cells. Nuclear staining for prelamin A (non farnesylated), emerin and lamin A/C was similar between T528M and control preadipocytes when adding pioglitazone. The signal of farnesylated prelamin A was mildly more intense in the nuclear periphery and cytoplasm of T528 cells mainly before differentiation (Figure 6).

## 4. Discussion

FPLD2 or Dunnigan disease is a rare condition due to variants in the *LMNA* gene, usually affecting codons 8 to 11. This disorder is characterized by the lack of subcutaneous fat in the lower and upper limbs, buttocks and abdomen, and the presence of insulin resistance with metabolic syndrome. We report five patients with the missense variant c.1583C>T, p.(Thr528Met) in exon 9 of *LMNA*, and various polymorphisms in *LMNA* and *PPARG*. Other variants in genes related to adipogenesis were ruled out.

To this day, the p.(Thr528Met) change in *LMNA* has been associated with FPLD and progeroid syndromes albeit only in compound heterozygosity [1,2]. Savage et al. [1] published the case of a female with FPLD2, who carried two *LMNA* missense variants: c.1748C>T (p.S583L) in exon 11, inherited from her father, and c.1583C>T, p.(Thr528Met) in exon 9, inherited from her mother. Relatives with only one of the changes did not exhibit features of lipodystrophy, while those with both of them had the typical FPLD2 phenotype. In 2006, Verstraeten et al. [2] reported a male affected by a progeroid syndrome with two *LMNA* missense variants: c.1619T>C, p.(Met540Thr) in exon 10, inherited from his mother, and c.1583C>T p.(Thr528Met) in exon 9, inherited from his father. Although both parents did not suffer from any apparent disease, histological and molecular investigations of their nuclei showed several abnormalities similar to those of the son, albeit at a much lower frequency. Additionally, another variant in the same codon of *LMNA*, c.1583C>G, p.(Thr528Arg), has been associated with Emery–Dreifuss muscular dystrophy [14]. Nucleotide changes in consecutive locus (c.1580G>A, p.(Ala527His); c.1585G>A, p.(Ala529Thr); and c.1586C>T, p.(Ala529Val)) lead to type A mandibuloacral dysplasia [15,16,17].

First, in this study, we show that the T528M variant could lead by itself to FPLD, but with a variable lipoatrophic phenotype, ranging from obvious to subtle forms, not as severe as in the case of exon 8 variants. It is unclear why earlier studies did not reach similar results, but it should be borne in mind that variable expressiveness and incomplete penetrance are common in Mendelian diseases. Other possibilities include the influence of the environment, which might induce epigenetic changes or the presence of single nucleotide polymorphisms (SNPs) in *LMNA* and/or other variants in other genes that could explain the differences and modulate the pathogenic effect of the T528M variant. Interestingly, the Pro12Ala polymorphism in *PPARG* found in Case #1 has been associated with an inhibitory effect in thiazolidinedione-induced adipogenesis [18]. On the other hand, the complications associated with the T528M variant do not seem so severe insofar as there was only one case with ischemic heart disease and another with steatosis, although it must be taken into account that the sample size is reduced. Hypertriglyceridemia was present in all patients, but not as severe as in R482W patients, as there were no cases of acute pancreatitis.

Secondly, cellular and gene expression studies support our findings in patients, that is, the genotype-phenotype segregation for this *LMNA* variant, and that therefore the T528M variant could alter adipogenesis. We found significant decreases in T528M cells for *CEBPA*, *LPL* and *FABP4* gene expressions. Lipoprotein lipase, which determines the free fatty acid deposition in white adipose tissue, was almost undetectable in T528M cells, as is the case of the adipocyte fatty acid binding protein 4, which is produced primarily by adipocytes and whose blockade enhances insulin sensitivity. Moreover, the reduction in *CEBPA* expression might indicate an alteration of the adipogenesis. Strikingly, *PPARG* expression was significantly higher in T528M cells, whereas its expression is reduced in the classic form of FPLD2 [19,20]. Recently, Oldenburg et al. (2017) demonstrated that the R482W variant in *LMNA* inhibits adipogenesis by epigenetically deregulating long-range enhancers of the anti-adipogenic miR335 microRNA gene in human adipocyte progenitor cells [21]. As it is well known that *PPARG* is the master of adipogenesis, we speculate that the T528M variant alters the adipogenic differentiation by different mechanisms and that increased *PPARG* expression might be the reflection of a compensatory operating system. If this change in *PPARG* expression could explain the not so severe lipoatrophic phenotype, it would be a matter of speculation.

Regarding immunofluorescence studies, the mild increase in farnesylated prelamin A in T528M cells might illustrate an intrinsic alteration in the lamin maturation, consistent with previous studies on the pathogenesis of FPLD2 [20]. Even if the recent hypothesis of the miRNA-related heterochromatin alterations is gaining more relevance, we have to take into account that this was demonstrated only for the R482W variant [21], and it cannot be ruled out that the accumulation of immature prelamin altering adipogenesis by sequestering certain transcription factors as has been shown by other studies [20,22,23], may play a role in the pathogenesis of lipodystrophy associated with the T528M variant. However, this finding should be taken with caution, and further experiments in others patients cell cultures would merit to be done in order to confirm this issue.

## 5. Conclusions

This work deepens the knowledge of laminopathies and lipodystrophies, and it stresses the amazing complexity of these pathologies: The same disease may occur via different variants and, in turn, the same variant can lead to different diseases. Although the results point to the direction that the T528M variant in *LMNA* could produce FPLD2 by itself with a lipoatrophic phenotype varying from subtle to evident forms, we consider that this study deserves further deepening.

## Figures and Tables

**Figure 1 jcm-10-01497-f001:**
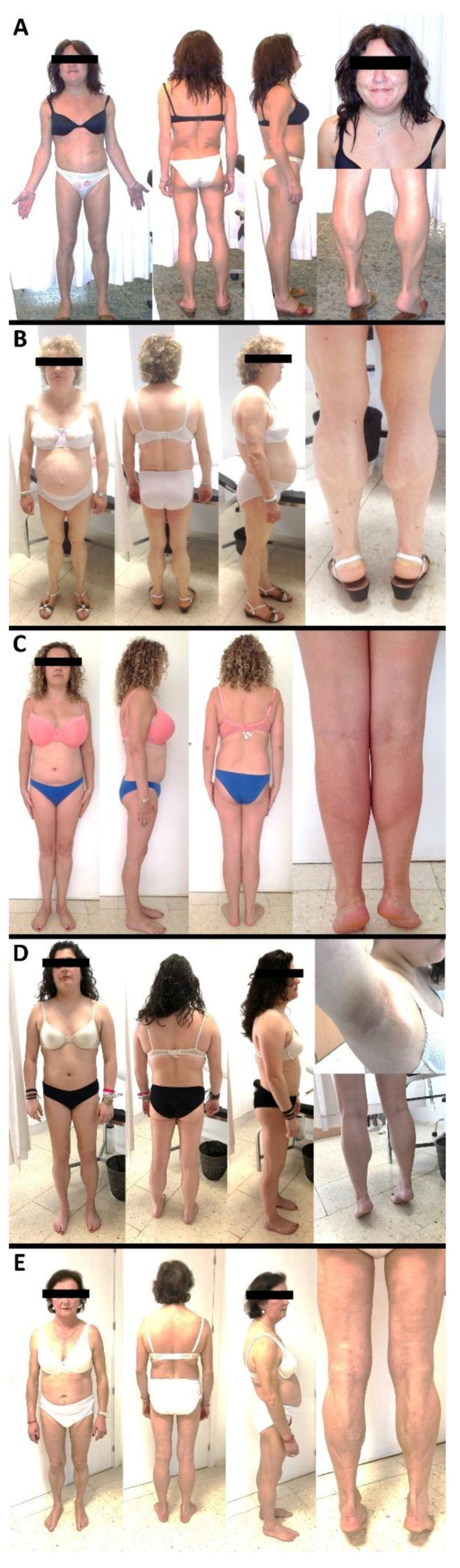
Photographs of the five cases with FPLD2 bearing the p.(Thr528Met) variant in the *LMNA* gene. The photographs show body morphology caused by the p.(Thr528Met) variant. (**A**) Case #1 is a 53-year-old female. Fat loss higher than in the other patients. Lack of subcutaneous adipose tissue in limbs, abdomen and buttocks. Rounded face, double chin, hypermuscular appearance with calf hypertrophy, hepatomegaly, phlebomegaly and small breasts. (**B**) Case #2 is a 58-year-old female. Fat loss in upper limbs, hips, thighs and calves. Fat accumulation in face with double chin, in upper back and intra-abdominal region. Scarce abdominal subcutaneous adipose panicle. Phlebomegaly. Calf hypertrophy. Acanthosis nigricans on the nape and axillae. Hepatomegaly, splenomegaly. (**C**) Case #3 is a 34-year-old female. No obvious fat loss. Low fat in arms, buttocks, hips and lower extremities. Fat accumulation in face, double chin, trunk, abdomen and axillae. Minimal acanthosis nigricans. (**D**) Case #4 is a 20-year-old female. Fat accumulation in face, double chin, dorsal region, axillae, neck, arms and scapular region. Scarce fat in hips, buttocks, thighs and calves. Phlebomegaly and marked musculature in lower limbs. Normal amount of fat in abdomen and upper limbs. Hirsutism. Acanthosis nigricans in axillae, nape and groin areas. (**E**) Case #5 is a 62-year-old female. Fat loss in upper limbs, lower limbs, buttocks and hips. Accumulation of fat in face, abdomen, chin, back and axillae. Well-defined muscles in arms, legs and buttocks. Minimal phlebomegaly in arms.

**Figure 2 jcm-10-01497-f002:**
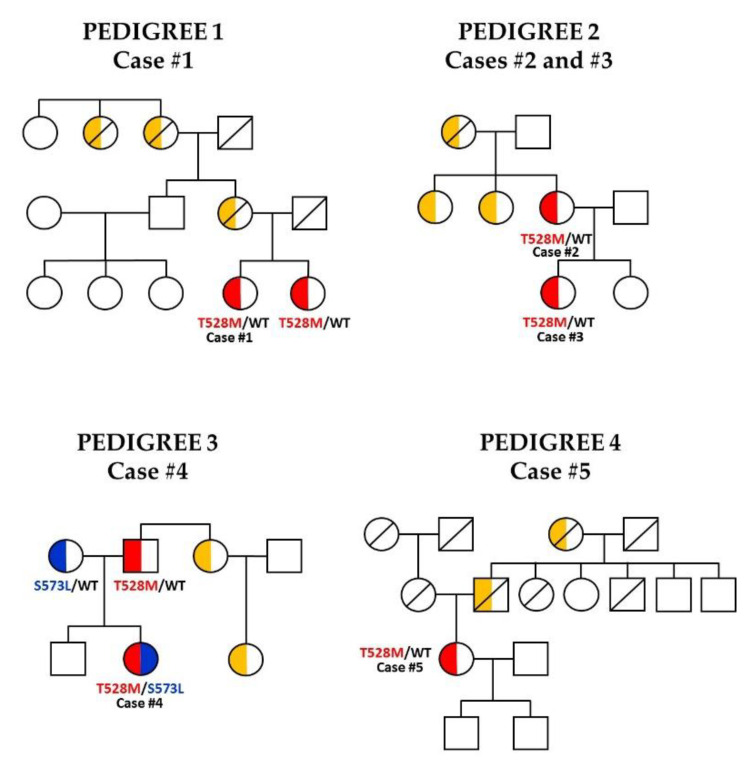
Pedigrees of the four families with FPLD2 due to the p.(Thr528Met) variant. Genograms. Affected individuals with the *LMNA* p.(Thr528Met) variant and the *LMNA* p.(Ser573Leu) variant are shown as half-filled red and blue symbols, respectively, unaffected subjects as unfilled symbols, individuals for whom the phenotype is suspected are shown as half-filled orange symbols. Squares denote males, and circles denote females. Circles and squares with a diagonal slash denote deceased subjects. The inheritance pattern was autosomal dominant in a vertical way.

**Figure 3 jcm-10-01497-f003:**
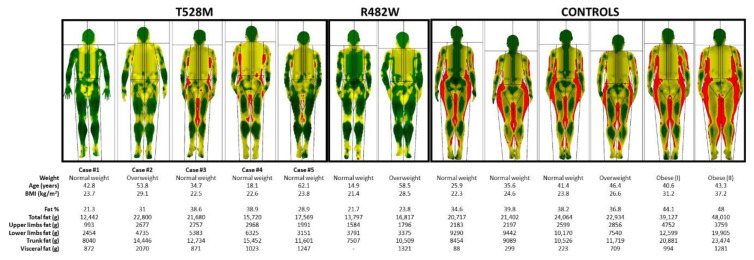
DXA scans of patients with the p.(Thr528Met) and p.(Arg482Trp) variants in the LMNA gene and control subjects. The total body scans were color-mapped, with green representing an area of low level % fat (0–25%), yellow an area of medium level % fat (25–60%) and red an area of high level % fat (60–100%).

**Figure 4 jcm-10-01497-f004:**
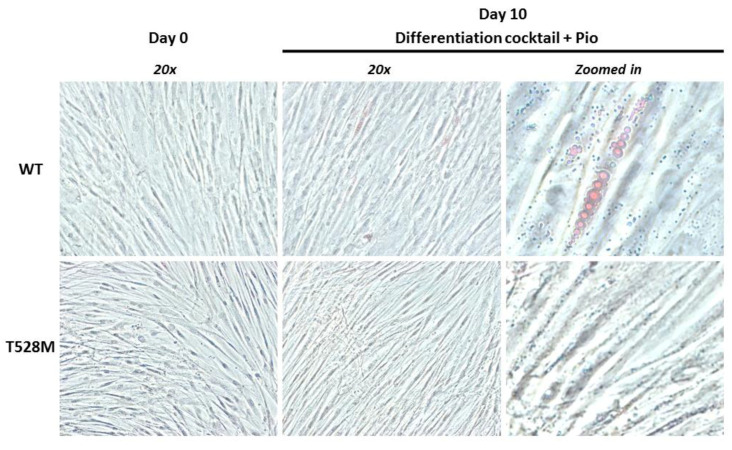
Representative photographs of lipid droplet formation in preadipocytes demonstrated by using Oil red O staining. Preadipocytes were induced for differentiation over 10 days into adipocytes with the differentiation cocktail containing pioglitazone. At day 10, after the induction of adipogenic differentiation, microphotographs were taken from a central area of a representative triplicate well. Phase contrast microscopy images were visualized under 20-fold magnification; 4-fold zoomed in areas are provided.

**Figure 5 jcm-10-01497-f005:**
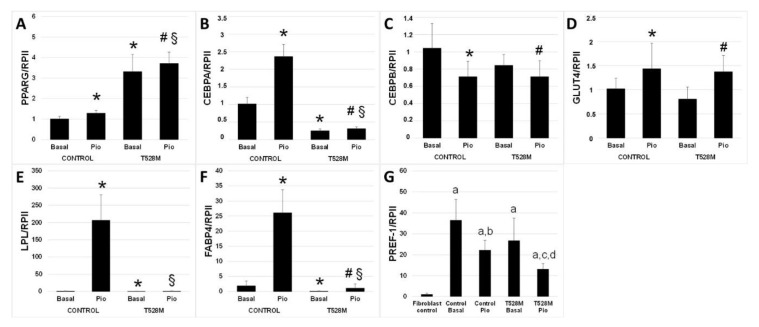
Relative expression of adipogenic genes. Relative expression of adipogenic genes: (**A**) *PPARG*, (**B**) *CEBPA*, (**C**) *CEBPB*, (**D**) *GLUT4*, (**E**) *LPL* and (**F**) *FABP4*; * *p* < 0.05 vs. Control, ^#^
*p* < 0.05 vs. T528M Basal, ^§^
*p* < 0.05 vs. Control Pio; and relative expression of the adipocyte marker (**G**) *PREF-1*, a: *p* < 0.001 vs. Fibroblast control, b: *p* < 0.05 vs. Control basal, c: *p* < 0.01 vs. T528M basal, d: *p* < 0.01 vs. Control Pio. All samples were analyzed in triplicate, *n* = 4. Results were normalized to the *RNA polymerase II* gene.

**Figure 6 jcm-10-01497-f006:**
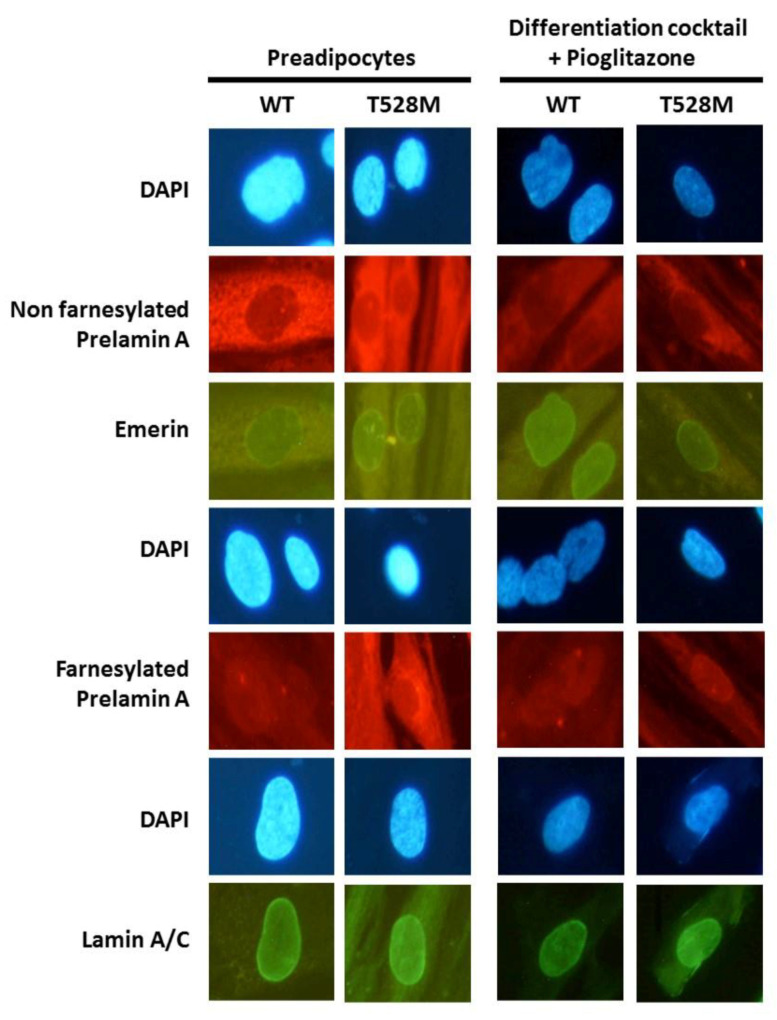
Immunofluorescence images of representative preadipocytes and differentiated adipocytes. All images were photographed at 20-fold magnification.

**Table 1 jcm-10-01497-t001:** Primer sequences and probes.

Genes	Forward Primer (5′–3′)	Reverse Primer (5′–3′)	Probe	Probe Sequences	Amplicon Length (nt)
***CEBPA***	GCAAATCGTGCCTTGTCAT	CTCATGGGGGTCTGCTGTAG	12	CTCCTTCC	72
***CEBPB***	CGCTTACCTCGGCTACCA	ACGAGGAGGACGTGGAGAG	74	CTGCTGCC	65
***FABP4***	CCTTTAAAAATACTGAGATTTCCTTCA	GGACACCCCCATCTAAGGTT	72	TTCCTGGC	105
***GLUT4***	CTGTGCCATCCTGATGACTG	CGTAGCTCATGGCTGGAACT	67	TGCTGGAG	62
***LPL***	ATGTGGCCCGGTTTATCA	CTGTATCCCAAGAGATGGACATT	25	CTCCTCCA	76
***PPARG***	GACCTGAAACTTCAAGAGTACCAAA	TGAGGCTTATTGTAGAGCTGAGTC	39	CTCCACCT	95
***PREF-1***	GACGGGGAGCTCTGTGATAG	CATAGAGGCCATCGTCCAG	68	AGGAGCAG	94
***RNA polymerase II***	GCATCATGAACAGCGATGAG	TCATCCATCTTGTCCACCAC	69	GGAGGAAG	64

**Table 2 jcm-10-01497-t002:** Demographic, anthropometric, biochemical and clinical features of the studied subjects.

Demographic Features					
	Case #1	Case #2	Case #3	Case #4	Case #5
Age	42.8	53.8	34.7	18.1	62.1
Sex	F	F	F	F	F
**Variants**					
*LMNA*	c.1583C>T p.(Thr528Met)	c.1583C>T p.(Thr528Met)	c.1583C>T p.(Thr528Met)	c.1583C>T p.(Thr528Met)/c.1718C>T p.(Ser573Leu)	c.1583C>T p.(Thr528Met)
*Other genes*	-	WRN c.1495A>G p.(Arg499Gly); BLM c.813G>C p.(Lys271Asn)	WRN c.1495A>G p.(Arg499Gly); BLM c.813G>C p.(Lys271Asn)	-	-
Autosomal dominant inheritance	yes	yes	yes	yes	yes
SNP *PPARG*p.Pro12Ala	yes	no	no	no	no
**Clinical features**					
Acanthosis	no	yes	yes	yes	no
Phlebomegaly	no	yes	no	yes	yes
Hypermuscularity	yes	yes	no	no	yes
Lipomas	no	no	yes	no	no
Goiter	no	no	no	no	yes
Diabetes mellitus (DM)	yes	yes	no	no (IFG)	no (IFG)
Dyslipidemia	IV	IIb	IV	IV	IIb
Steatosis	no	yes	no	no	no
Arterial hypertension (AHT)	yes	yes	no	no	yes
Cardiovascular diseases (CVDs)	no	no	no	no	yes
Polycystic ovary syndrome (PCOS) and obstetric complications	no	no	no	yes	no
Pancreatitis	no	no	no	no	no
Lipodystrophy onset	Childhood	Childhood	31 years of age	Adolescence	Adolescence
Family background	Mother and sister: FPLD	Daughter: FPLD; Mother: DM	Mother: FPLD	Father: hypertriglyceridemia	Mother: DM
**Anthropometric data**					
Weight (kg)	58.5	73.5	58.2	61.5	60.8
Height (cm)	157	159	161	165	160
BMI (kg/m^2^)	23.7	29.1	22.5	22.6	23.8
Waist (cm)	76	101	84	84	90
Hip (cm)	87	100	87	94	89
Waist-to-height ratio	0.9	1	1	0.9	1
Waist-to-hip ratio	0.48	0.64	0.52	0.51	0.56
**Skinfold thickness (mm)**					
Triceps	5.5	5	16	20	6
Biceps	4	7	8	11	5
Suprailiac	9	18	23	24	11
Subscapular	17.5	34	28	52	28
Thigh	3.9	6	15	12	4
Calf	3.7	3	15	15	2
**DXA scan, fat mass (g)**					
Fat %	21.3	31	38.6	38.9	28.9
Total fat	12,442	22,800	21,680	15,720	17,569
Upper-limb fat	993	2677	2757	2968	1991
Lower-limb fat	2454	4735	5383	6325	3151
Trunk fat	8040	14,446	12,734	15,452	11,601
Visceral fat	872	2070	871	1023	1247
**Biochemical features**					
Basal glucose (mg/dL)	103	218	67	94	96
Hemoglobin A1c (%)	6.2	7.2	5.3	5,.2	5.7
Plasma insulin (mIU/l)	5.9	ND	20	57.8	13
Peptide C (ng/mL)	ND	2.1	ND	3,4	2,2
Plasma leptin (ng/mL)	2.4	9.7	ND	13	3,3
Total cholesterol (mg/dL)	146	174	264	220	161
Plasma triglycerides (mg/dL)	372	247	214	338	151
High-density lipoprotein cholesterol (HDLc) (mg/dL)	26	36	53	39	45
Low-density lipoprotein cholesterol (LDLc) (mg/dL)	39	89	169	113	86
Alanine aminotransferase (ALT) (IU/L)	27	56	13	44	26
Aspartate aminotransferase (AST) (IU/L)	16	65	16	26	20
Gamma-glutamyl transpeptidase (GGT) (UI/L)	9	57	23	23	14
Creatinine (mg/dL)	0.81	0.5	0.5	0.59	0.64
Creatine kinase (CK)	ND	103	54	81	110
Thyroid stimulating hormone (TSH)	1.43	2.02	2.78	3.27	3.58
Blood pressure (BP)	136/79	147/80	125/69	135/89	155/87
Echocardiogram (ECHO)	-	normal	-	normal	mitral regurgitation
Medication	Sitagliptin, Fenofibrate, Omega-3 fatty acids, Telmisartan	Metformin, Insulin, Ramipril, Rosuvastatin, Aspirin, Dapaglifozin	-	Metformin	Lormetazepam, Clopidogrel, Aldactone, Atorvastatin

“Obstetric complications” include miscarriages, gestational diabetes and/or macrosomy. IFG, impaired fasting glucose. ND, not determined. SNP, single nucleotide polymorphisms.

## Data Availability

The data presented in this study are available within the article.

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
