# Peer review of "Variable Expressivity and Allelic Heterogeneity in Type 2 Familial Partial Lipodystrophy: The p.(Thr528Met) LMNA Variant"

_jcm, 2021, doi:10.3390/jcm10071497_

Round 1

Reviewer 1 Report

Variable expressivity and allelic heterogeneity in type 2 familial partial lipodystrophy: the p.(Thr528Met) LMNA variant

Summary:

The aim if the study is to establish that missense variant in exon9 of LMNA gene is the primary cause of the familial partial lipodystrophy and this mutation alters adipogenesis in the preadipocytes obtained from humans. The paper also focuses on genotype-phenotype associations in FPLD patients. To accomplish this, 5 FPLD patients carrying c1583C>T, p(Thr528Met) variant in exon 9 of the LMNA gene were evaluated. Dual energy X rat absorption was used for the assessment of anthropometric features and body composition. Family history, physical and biochemical examination was done to get some in-dept background knowledge. Intracellular localization of lamins were studied using real time PCR and immunofluorescence. Based on the experiments, the authors conclude that T528M variant in LMNA gene leads to altered adipogenic machinery in FPLD2 patients wherein the preadipocytes fail to differentiate, reduced expression of adipogenic genes and accumulation of farnesylated prelamin A in the T528M cells.

Review:

Overall, the phenotypic analysis and clinical characterization of the patients is detailed and well performed. The fundamental hypothesis of this paper, which is that p.T528M variant in LMNA is the causal mutation is not substantiated due to major issues in the cellular analysis.

Major:

In Line 312 the author claims that the Case1 preadipocytes do not differentiate when exposed to differentiation cocktail. This is true but the WT control cells do no differentiate either as shown by the lack of Oil red O staining in control adipocytes after treatment with differential cocktail. Thus no conclusion can be drawn as no differentiation is observed in any cells mutant or control.

In line 344, According to the authors, the signal for farnesylated prelamin A was mildly higher in the nuclei of p.T528M cells before and after differentiation. In figure 6 the signal for farnesylated prelamin A cannot be seen clearly due to the poor quality of the images and overexposure in column 2.

Minor:

The SNP analysis is not clearly rationalized. The author used PPARG exons 1 and 7 and LMNA exons 7,7 and 10. It is unclear why only these specific exons were used. And these SNPs are common and would be expected to have no relationship to a Mendelian disease such as FPLD2.

In line 321. Figure 5A-G needs to be replaced by Figure 5A-F.

In line 322. Figure 5H needs to be replaced by Figure 5G.

In line 337. (H) PREF-1 needs to be replaced by (G) PREF-1

Reviewer 2 Report

This is an interesting case series of 5 FPLD2 patients. The authors aimed to explore whether the p.(Thr528Met) LMNA variant which was common in all 5 cases, was responsible for the observed phenotype. Adipocytes from one of these cases were studied, to assess for possible defects in adipogenesis and the genotype-phenotype associations were examined.

Comments:

  1. In vitro adipocyte studies revealing impaired differentiation and reduced expression of adipogenic genes, were performed for case #1 only. It is therefore possible that these results are not generalizable for all patients carrying the T528M variant. For example, as stated by the authors, the Pro12Ala polymorphism in PPARG found in this particular case has been associated with an inhibitory effect in thiazolidinedione-induced adipogenesis in some (Ref14), albeit not all (Kamble PG et al, https://doi.org/10.1080/21623945.2018.1503030) studies.
  2. In case #3 the phenotype is indeed, very mild (as stated in the manuscript). In fact, the patient has no clear metabolic abnormalities (apart from mild hypertriglyceridemia and "android" fat distribution) and she is not requiring any therapy. In other words, the presence of the T528M variant in LMNA may not always lead to a clear FPLD2 phenotype (additional factors might be necessary, as indicated by the authors in the discussion, lines 372-379). It may therefore be more accurate to state in the conclusion (in the main manuscript and in the abstract) that the T528 variant in LMNA may lead to FPLD2.
  3. page 1, line 28, abstract: you may wish to substitute the word "completed" by "recorded"
  4.  page 2, line 71: please include an explanation for the NGS abbreviation (presumably, "next-generation sequencing")
  5. Table 2, line 213: please remove the DOB (date of birth)
  6. Table 3, line 310. Please describe in the legend that the table refers to the analysis of SNPs from a cohort of 62 FPLD2 patients and 42 controls. In addition, you may better explain in the text (line 309) why you are presenting these data.

Reviewer 3 Report

This paper presents some interesting case reports of FPLD patients with different genetic variants. Genetic and clinical heterogeneity in laminopathies are really a big deal and trying to correlate extensive phenotypic observations with genetic variants description is always interesting.

My major concern is about genetic description and interpretation of variants. Although correct nomenclature is given, it should be interesting to get an idea of the biological interpretation of these variants, ie to use the ACMG guidelines to classify them (Richards & al. 2015).

Indeed, the single use of Sift and Polyphen is old fashioned and some powerful tools now exist to collect data on variants, such as Clinvar, gnomAD, Varsome, Mobidetails… In pathology, variants are essentially retained by their frequency, not anymore regarding pathogenicity prediction with bioinformatics tools…

So this section on variants could be greatly improved.

Minor concerns

  • L48 : lamins C2 and Delta10 are non-ubiquitous, but specific to germline cells
  • L66 : lamin C is directly translated from the messenger RNA in its mature soluble form
  • Table 1 : please, indicate the 5’-3’ orientation of primers
  • Table 2 : please, indicate all the units (weight, height, biochemistry…) several are missing
  • Fig 2 : I personally think that writing T528T instead of “WT” is more difficult to understand, thus variants are more unclearly visible
  • L236 : what about the affected sister ? has she been genotyped ? if so, please complete Figure 2 appropriately, or add a sentence to explain…
  • L296 : this sentence is hard to understand. Why not simply say she bears another variant ?
  • L295-304 : please add at least gnomAD frequencies, Clinvar reports and ACMG variant classification. For instance, PLIN1 variant is reported more than 30 000 times in gnomAD, with a lot of homozygous, indicating more strongly than bioinformatics predictions that it is a polymorphism (highly probably)
  • Table 3 : please comment this table. is there a statistical analysis available ? what do you want to enlighten ?
  • Fig 4 : could you please point (with arrows or another sign) what is to be seen ? it is extremely difficult to observe the signal on this pictures
  • Fig 5 : are you sure the statistical signs are correctly positioned ? especially # ?
  • Fig 5H is missing
  • Fig 6 : I’m not sure the signal obtained with farnesylated/non-farnesylated prelamin A is specific ? at least, farnesylated protein should colocalize with membranes, either RE or nuclear envelope… it seems to spread everywhere…what would you like to demonstrate with these observations ?

Round 2

Reviewer 1 Report

In the author revision our comments were:

1) Poor differentiation in the WT control: The response provided magnified a single differentiated cell in the field. This image further confirms the lack of differentiation in the entire experiments as now it is even more clear that the rest of the cells are no differentiated. The qPCR results are mixed and do not tell a clear story of decreased differentiation in the mutant cells. Pref-1 a preadipocyte marker decreases in both the mutant and WT cell upon differentiation.

2) Regarding the Farnesylated lamins, the new images provided do not have the magnification or resolution to enable discernment of any difference. For reference please see figure 3 in the paper linked by the url (https://jcs.biologists.org/content/119/16/3265) for the type of imaging required to support the claim 

Ultimately, the p.T528M cellular data are inconclusive and do not support a conclusion of a causal mutation and so we would recommend tempering this claim in the manuscript.
